# Characterizing Spatiotemporal Patterns of Land Subsidence after the South-to-North Water Diversion Project Based on Sentinel-1 InSAR Observations in the Eastern Beijing Plain

Yuanyuan Liu [1,2,3], Xia Yan [1], Yuanping Xia [1,2], Bo Liu [1,2,*], Zhong Lu [4] and Mei Yu [1,2]

1. Faculty of Geomatics, East China University of Technology, Nanchang 330013, China
2. Key Laboratory of Mine Environmental Monitoring and Improving around Poyang Lake, Ministry of Natural Resources, Nanchang 330013, China
3. Key Laboratory for Digital Land and Resources of Jiangxi Province, East China University of Technology, Nanchang 330013, China
4. Roy M. Huffington Department of Earth Sciences, Southern Methodist University, Dallas, TX 75275, USA
* Correspondence: liubo@ecut.edu.cn

**Abstract:** The eastern Beijing plain has been suffering severe subsidence for the last decades, mainly associated with the long-term excessive extraction of groundwater resource. Since the end of 2014, the annual water supply in Beijing plain has reached several hundred million cubic meters because of the South-to-North Water Diversion (SNWD) Project, which has reduced the groundwater exploitation and changed the status of land subsidence. In this work, we first obtain the current spatiotemporal variations of land subsidence in the eastern Beijing plain by using progressive small baseline subsets (SBAS) InSAR time series analysis method with Sentinel-1 SAR data acquired from July 2015 to December 2021. Then, we analyze the correlations between InSAR-derived subsidence and groundwater level change by applying the cross wavelet method. The results show that two major subsidence zones are successfully detected with the maximum deformation rate of −150 mm/yr and maximum cumulative deformation of −950 mm. Besides, the ground deformation at different stages from 2016 to 2021 reveal that the area and magnitude of major deformation significantly slow down, even in the regions with severe subsidence, especially in the year of 2017, which is about two years later than the start time of SNWD Project in Beijing. Further, we find the InSAR-derived subsidence lags groundwater level change with about 1–2-month lagging time, indicating that the dynamic variation of groundwater level fluctuation may be the main factor affecting the uneven subsidence in the severe subsiding zones. Last, differential subsidence rates are identified at both sides of geological faults, such as Nankou-Sunhe fault and Nanyuan-Tongxian fault, from the observed deformation map, which could be explained that the groundwater flow is blocked when a fault is encountered. These findings can provide significant information to reveal the deformation mechanisms of land subsidence, establish the hydrogeological models and assist decision-making, early warning and hazard relief in Beijing, China.

**Keywords:** South-to-North Water Diversion (SNWD) Project; InSAR; land subsidence; cross wavelet method; Sentinel-1; Beijing plain

## 1. Introduction

As a common environmental geological phenomenon, land subsidence refers to the decline of regional ground elevation owing to the influence of natural factors or human engineering activities [1]. With long-term over-exploitation of groundwater and rapid urban expansion, land subsidence has become a worldwide widespread geological hazard, such as Mexico [2–4], Houston [5], Willcox Basin, Arizona [6], Northern Italy [7], Semarang, Indonesia [8]. In China, many cities, including Beijing [9–16], Shanghai [17], Xi'an [18], Wuhan [19] and Taiyuan [20], are suffering from the potential land subsidence.

In order to alleviate and mitigate the disasters related to land subsidence, it is essential to characterize the spatiotemporal patterns of land subsidence with high precision. Several ground-based geodetic surveying techniques (e.g., GNSS, leveling) have been applied to obtain the ground displacements in the typical regions. Because these techniques are time-consuming, labor-intensive and only have a few sparsely distributed monitoring stations, the applications are limited in long-term and large-scale deformation monitoring. Since the end of the 20th century, Synthetic Aperture Radar Interferometry (InSAR) technique has been an effective geodetic remote sensing tool for measuring large-scale deformation from space, with the advantages of large space coverage, high spatial resolution and millimeter-scale accuracy [21] and has been successfully used in many areas [2–20].

However, there are also several following challenges for retrieving the high-precision ground deformation with conventional InSAR technique: spatial or geometrical decorrelation [22], temporal decorrelation [22] and atmospheric disturbances [23]. Subsequently, a series of advanced InSAR time series analysis methods based on multi-interferograms have been proposed and developed by many research institutions to obtain the deformation with high precision [24–30]. Regarding the deformation monitoring of land subsidence in Beijing with InSAR observations, several related results with ERS-1/2, ENVISAT ASAR, ALOS PALSAR-1/2, RADARSAT-2, TerraSAR-X and Sentienl-1 SAR datasets have been issued in recent years [9–16,31–44], which has shown that land subsidence has been developed in the eastern Beijing plain for the last decades (e.g., 2003–2015, 2015–2017) with large deformation rate (>100 mm/yr) and is mainly caused by the excessive extraction of groundwater resource and the construction of urban buildings and infrastructures [31]. However, the middle route of the South-to-North Water Diversion (SNWD) Project, which starts from Danjiangkou Reservoir and ends in Beijing, has reduced the groundwater exploitation and altered the water consumption structure of Beijing plain since December 2014, changing the status of land subsidence [43]. Unfortunately, few studies are published concerning the correlations between SNWD Project and land subsidence in Beijing plain [40,45]. Therefore, the present development, more detailed spatiotemporal changes of land subsidence after the SNWD Project should be answered in the eastern Beijing plain.

The main objective of this work is to reveal the latest spatiotemporal changes of land subsidence over the eastern Beijing plain and assess the roles that groundwater exploitation, urban construction and geological faults played in the deformation. Firstly, a progressive SBAS-InSAR approach was adopted to obtain the current deformation rate and deformation time series in the eastern Beijing plain (2015–2021) by using 72 C-band Sentinel-1 SAR images. Then, we combined the InSAR-derived subsidence and groundwater level measurements to reveal the potential relationship and lagging time between the InSAR-derived subsidence and groundwater level changes. Finally, the possible links between the InSAR-derived subsidence, urban expansion and geological faults are discussed.

## 2. Study Area and Dataset

### 2.1. Study Area

Beijing, the capital of China, covers an area of about 16,412 km$^2$. It is surrounded by mountains in the west, north and northeast and the average ground elevation is approximately 43.5 m. As shown in Figure 1, the study area of this paper is situated in the eastern part of Beijing plain, which presents a typical semi-humid and semi-arid continental monsoon climate. The average annual temperature is about 11–12 °C [15]. The seasonal rainfall is unevenly distributed, and approximately 80% of the total annual precipitation concentrates between mid-June and September [16]. Furthermore, this region is a typical piedmont alluvial diluvial plain composed of sediments from three major river systems, namely Yongding, Wenyu and Chaobai Rivers. Based on the stratigraphic age, genesis type and burial depth, the Quaternary aquifers are characterized by a multi-layer aquifer system in our study area (Table 1). At present, the groundwater withdrawal is mainly from the Quaternary aquifer within the burial depth of 300 m [9]. For more detailed informa-

tion on the spatial extension of the aquifer systems in our study area, readers can refer to [40,44].

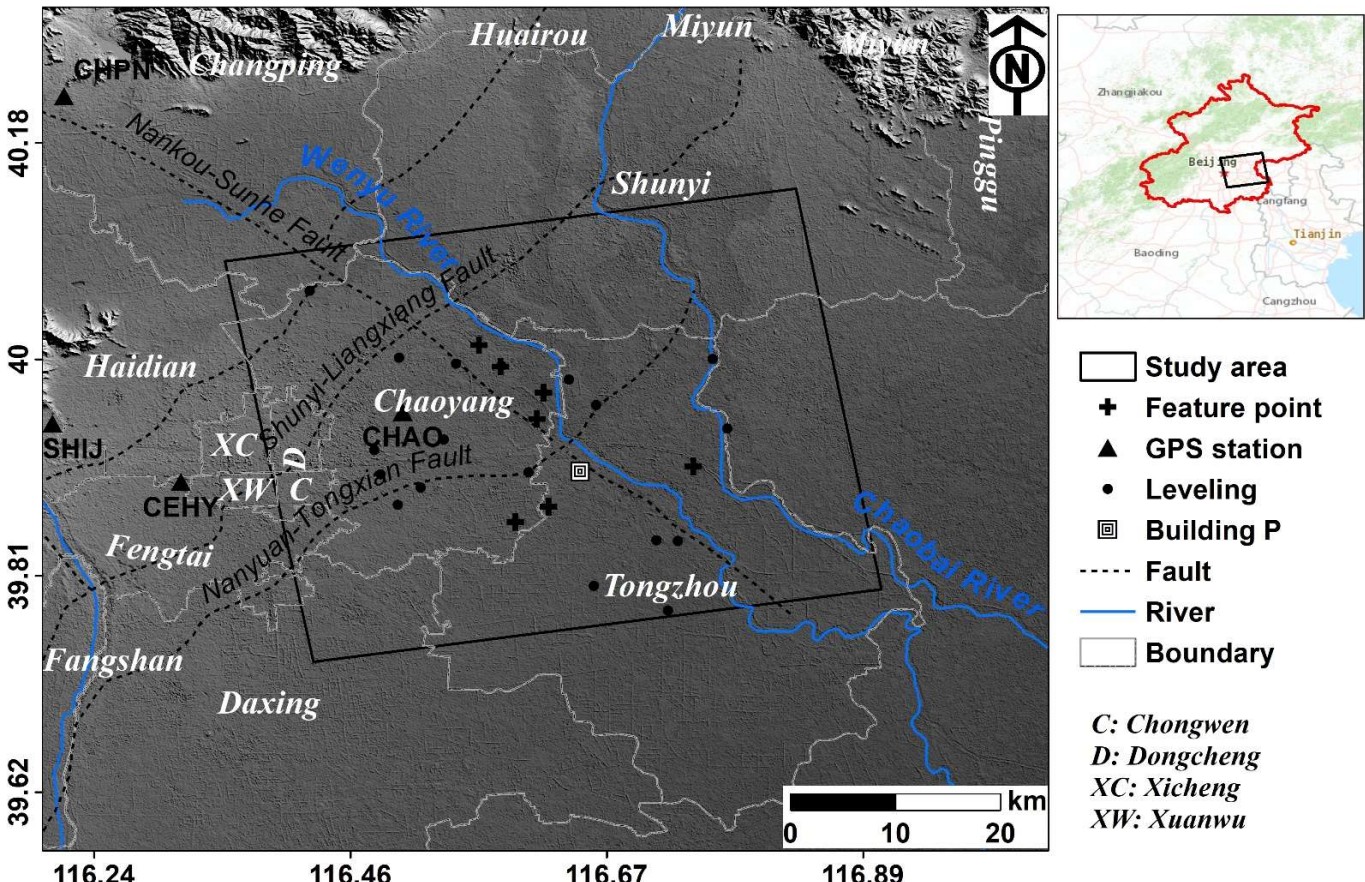

**Figure 1.** Shaded relief map of the eastern Beijing plain, where major rivers (blue solid lines), main active faults (black dashed lines), GPS stations (black triangles), leveling benchmarks (black dots) and the county boundary of Beijing administrative district (gray dashed lines) are all represented. The red polygon shown in the inset on the top-right corner indicates the location of Beijing and the black box illustrates the study area in this paper.

**Table 1.** The detailed information of aquifer groups in the study area.

| Aquifer Group | Main Lithological Features | Burial Depth (m) |
|---|---|---|
| The unconfined aquifer (I) | silt, silty sand and sandy clay | 0~50 |
| The first confined aquifer (II) | multiple types of gravel, sand and clay soil | 80~100 |
| The second confined aquifer (III) | multiple types of gravel, sand and clay soil | 100~180 |
| The third confined aquifer (IV) | mainly sand | 180~300 |

There are several main active geological faults cross throughout the study area, including Nankou-Sunhe fault, Nanyuan-Tongxian fault and Shunyi-Liangxiang fault. Nankou-sunhe fault is a NW-trending left-lateral normal fault, with the dip angle of 60°~68°, while Nanyuan-tongxian fault is a NNE-trending normal fault, with the dip angel of 40°~60°. According to the results from the Beijing earthquake station, the latest activity era of Nankou-Sunhe fault and Nanyuan-Tongxian fault are late Pleistocene and Holocene, whose mean slip rates are about 0.3 mm/a and 0.75 mm/a, respectively [46]. For more detailed information of the active geological faults, readers can refer to [46].

### 2.2. Dataset

In this work, the data set includes SAR images, GPS observations and leveling measurements (for the validation of InSAR-derived results). 72 C-band IW (Interferometric Wide) Sentinel-1 SAR images acquired from 30 July 2015 to 15 December 2021 are collected to reveal the latest land subsidence after the South-to-North Water Diversion (SNWD) Project in the eastern part of Beijing plain. For the progressive SBAS-InSAR processing, we divide the SAR data into two different groups. Among which, the first group refers to the archived SAR images (i.e., the first 56 SAR images acquired from 30 July 2015 to 24 December 2019) to generate possible differential interferograms (shown in black solid lines in Figure 2) with traditional SBAS-InSAR technique for deformation parameter initialization. The second group is the newly acquired SAR data to connect images with relatively new acquisition dates in the archived SAR data (i.e., the 55th and 56th SAR images in the first group) and generate new differential interferograms under the same spatial and temporal thresholds (shown in black dashed lines in Figure 2) to update new deformation parameters. In this study, the spatial and temporal baseline thresholds are set as 150 m and 96 days, respectively. GPS data were obtained from the GPS monitoring stations, which were measured by the China Earthquake Administration within the period of August 2015–December 2017. The measurements of 18 leveling benchmarks were measured by the Beijing Institute of Geological & Prospecting Engineering in China within the year of 2017. Meanwhile, the corresponding precise orbit ephemeris are obtained to correct the orbit information. Furthermore, the digital elevation model (DEM) data with a resolution of 30 m obtained from Shuttle Radar Topography Mission-1 (SRTM) is used to remove the topographic phases and geocode the deformation results.

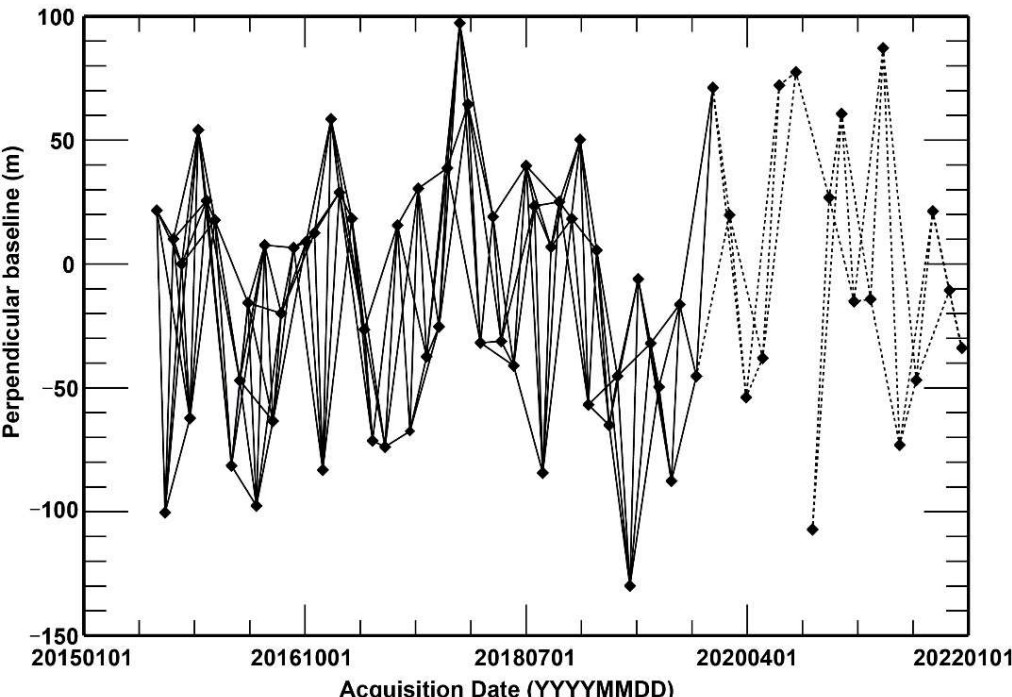

**Figure 2.** The spatiotemporal baseline distribution of differential interferograms used in this study. The black dots denote the acquisition dates of SAR data. The black solid lines represent the interferometric combinations of the archived SAR data in the first group for parameter initialization. The black dashed lines represent the new generated interferometric combinations between the newly acquired SAR data and relatively new acquisition dates in the archived SAR data for the sequential estimation.

## 3. Methodology

### 3.1. Progressive SBAS-InSAR Technique

In this paper, the deformation time series is calculated by using the progressive small baseline subsets (SBAS) InSAR approach. As shown in Figure 3, the data processing of this approach includes two major parts: (I) calculating preliminary time series of archived SAR images using traditional SBAS-InSAR technique and (II) updating deformation time series corresponding to the new SAR acquisition date with sequential estimation.

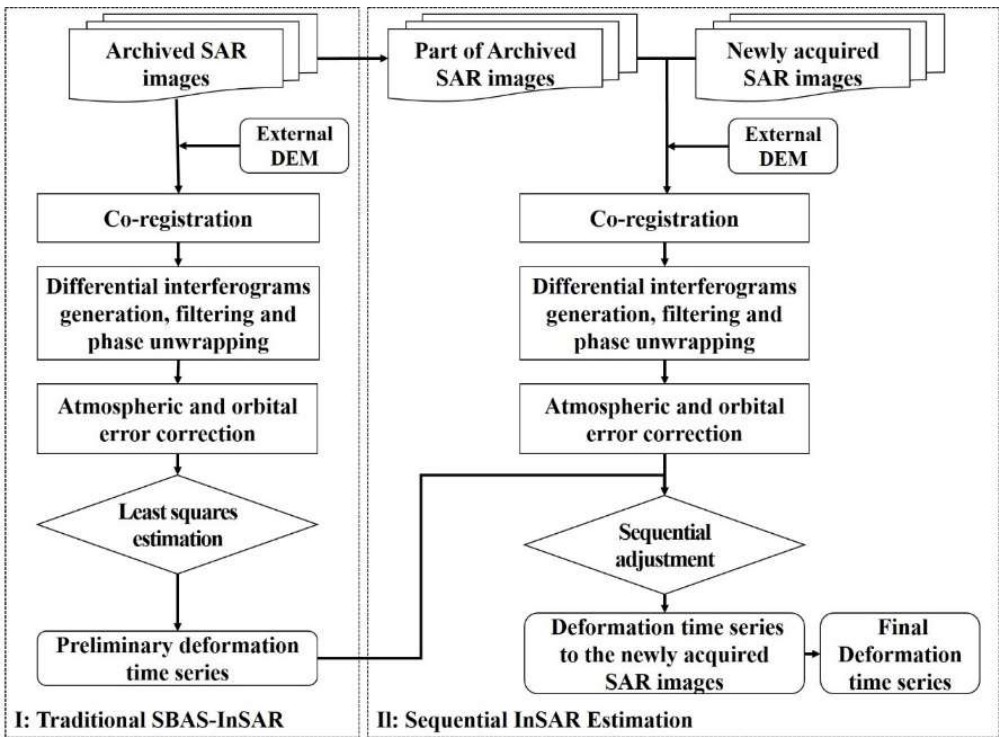

**Figure 3.** Flowchart of progressive SBAS-InSAR processing.

#### 3.1.1. Traditional SBAS-InSAR

Preliminary deformation time series of archived SAR images is calculated with the small baseline subsets (SBAS) InSAR technique [27,47] under the frame of GAMMA software. Considering $N + 1$ archived SAR acquisitions in chronological order $(t_0, t_1, \cdots, t_N)$, thus $M$ interferometric combinations can be obtained by setting the appropriate spatiotemporal baseline thresholds ($t_c$ and $b_c$). After removing topography-related phase and other phases induced by atmospheric disturbances and inaccurate orbital information from the interferograms, the *j-th* unwrapped differential interferogram $\delta\varphi_j$ generated between $t_A$ and $t_B$ can be expressed:

$$\sum_{k=t_A}^{t_{B-1}} v_{k+1}(t_{k+1} - t_k) = \delta\varphi_j \tag{1}$$

where $k$ is the acquisition time indexes, which ranges from $t_A$ to $t_{B-1}$; $v_k$ denotes the unknown deformation phase rate between the time-adjacent acquisitions.

Based on Equation (1), we can successfully obtain the deformation phase rates with the least squares (LS) method. Subsequently, under the assumption that the deformation at the first SAR acquisition is 0, we retrieve the cumulative deformation phases through an additional integration operation. Finally, we convert the deformation phases to ground displacements.

### 3.1.2. Sequential InSAR Estimation

Unlike traditional SBAS-InSAR, in which the cumulative deformation is re-estimated by using all SAR acquisitions when one or more new SAR images are added, we adopt the sequential adjustment to dynamically update the deformation parameters by just using the unwrapped interferograms obtained from those newly acquired SAR images. Assuming $N_1$ newly SAR acquisitions in chronological order ($t_{N+1}, t_{N+2}, \cdots, t_{N+N1}$), $M_1$ interferograms are generated under the same temporal and spatial baseline constraints, part of which are obtained by combining the newly acquired SAR data and $n$ images with relatively new acquisition dates (after $t_{n1}$) in the archived SAR data. Accordingly, the function model can be expressed as:

$$
\underbrace{\begin{bmatrix} t_1 - t_0 & 0 & \cdots & 0 \\ t_1 - t_0 & t_2 - t_1 & \cdots & 0 \\ \vdots & \vdots & \vdots & \vdots \\ 0 & 0 & \cdots & t_{n1-1} - t_{n1-2} \end{bmatrix}}_{A} \underbrace{\begin{bmatrix} v_1 \\ v_2 \\ \vdots \\ v_{t_{n1-1}} \end{bmatrix}}_{X_1} + \underbrace{\begin{bmatrix} t_{n1} - t_{n1-1} & 0 & \cdots & 0 \\ t_{n1} - t_{n1-1} & t_{n1+1} - t_{n1} & \cdots & 0 \\ \vdots & \vdots & \vdots & \vdots \\ 0 & 0 & \cdots & t_N - t_{N-1} \end{bmatrix}}_{B_1} \underbrace{\begin{bmatrix} v_{t_{n1}} \\ v_{t_{n1+1}} \\ \vdots \\ v_{t_N} \end{bmatrix}}_{X_2} = \underbrace{\begin{bmatrix} \delta\varphi_1 \\ \delta\varphi_2 \\ \vdots \\ \delta\varphi_M \end{bmatrix}}_{L_1} \tag{2}
$$

$$
\underbrace{\begin{bmatrix} 0 & 0 & \cdots & 0 \\ 0 & t_{n1+1} - t_{n1} & \cdots & 0 \\ \vdots & \vdots & \vdots & \vdots \\ 0 & 0 & \cdots & t_N - t_{N-1} \end{bmatrix}}_{B_2} \underbrace{\begin{bmatrix} v_{t_{n1}} \\ v_{t_{n1+1}} \\ \vdots \\ v_{t_N} \end{bmatrix}}_{X_2} + \underbrace{\begin{bmatrix} t_{N+1} - t_N & 0 & \cdots & 0 \\ t_{N+1} - t_N & t_{N+2} - t_{N+1} & \cdots & 0 \\ \vdots & \vdots & \vdots & \vdots \\ 0 & 0 & \cdots & t_{N+N_1} - t_{N+N_1-1} \end{bmatrix}}_{C} \underbrace{\begin{bmatrix} v_{t_{N+1}} \\ v_{t_{N+2}} \\ \vdots \\ v_{t_{N+N_1}} \end{bmatrix}}_{X_3} = \underbrace{\begin{bmatrix} \delta\varphi_{M+1} \\ \delta\varphi_{M+2} \\ \vdots \\ \delta\varphi_{M+M_1} \end{bmatrix}}_{L_2} \tag{3}
$$

where $A, B_1, B_2, C$ denotes the design matrix with the element of 0 or the interval between the time-adjacent acquisitions; $X_1, X_2, X_3$ denotes the unknown parameters associated with the deformation phase rates. It's worth noting that the initial solutions ($X_1'$ and $X_2'$) of $X_1$ and $X_2$ have been obtained from traditional SBAS-InSAR technique. Thus, $X_1$ and $X_2$ can be rewritten as:

$$
\begin{aligned}
X_1 &= X_1' + \delta X_1 \\
X_2 &= X_2' + \delta X_2
\end{aligned} \tag{4}
$$

where $\delta X_1$ and $\delta X_2$ represents the corrections of $X_1$ and $X_2$.

Assuming that all SAR images are mutually independent and the weight matrix equals to 1, the unknown parameters can be inversed by using sequential adjustment [48]:

$$
\begin{aligned}
\begin{bmatrix} \delta X_2 \\ X_3 \end{bmatrix} &= \begin{bmatrix} Q + B_2^T B_2 & B_2^T C \\ C^T B_2 & C^T C \end{bmatrix}^{-1} \begin{bmatrix} B_2^T L_2' \\ C^T L_2' \end{bmatrix} \\
Q &= B_1^T B_1 - B_1^T A (A^T A)^{-1} A^T B_1 \\
\delta X_1 &= -(A^T A)^{-1} A^T B_1 \delta X_2 \\
L_2' &= L_2 - B_2 X_2'
\end{aligned} \tag{5}
$$

where $Q$ represents the cofactor matrix derived from traditional SBAS-InSAR technique.

According to Equations (4) and (5), the updated deformation phase rates are obtained and the updated cumulative deformation phases are then solved through an additional integration operation. Therefore, the deformation parameters can be updated as quickly as possible once the new SAR images are involved.

### 3.2. Wavelet Transform for Time Series Analysis

Wavelet transform, one of the common time-frequency analysis methods, has been widely applied in various SAR/InSAR applications, especially in the analysis and interpretation of SAR data [15,49–51]. Cross wavelet transform (XWT) [52], which is a development of continuous wavelet transform (CWT), is used to explore the correlation between

two different time series (i.e., the InSAR-derived time series deformation and hydraulic level) across time and scales. In this work, the complex Morlet wavelet is selected to serve as mother wavelet in the processing. Assuming $W^X$ and $W^Y$ are the CWT results of two different time series $a(t)$ and $b(t)$, the cross wavelet transform $W^{XY}$ can be expressed as:

$$W^{XY} = W^X \cdot W^{Y*} \tag{6}$$

where $*$ denotes the complex conjugation.

Generally speaking, the product of the XWT is a complex number; therefore, the cross wavelet power spectrum ($|W^{XY}|$) is defined to reveal the correlation between $a(t)$ and $b(t)$, exposing the regions where the common power is high in the time-frequency domain [52].

Furthermore, the wavelet coherence (WTC) is introduced, which reveals the correlation between $W^X$ and $W^Y$ in the time-frequency domain. The wavelet coherence can be calculated as:

$$R^2(s) = \frac{\left|S(s^{-1}W^{XY}(s))\right|^2}{S\left(s^{-1}|W^X(s)|^2\right)S\left((s^{-1}|W^Y(s)|^2\right)} \tag{7}$$

where $R_n^2(s)$ denotes the wavelet coherence; $s$ denotes the scale of wavelet transform; $S$ is a smoothing operator. The value of $R_n^2(s)$ ranges from 0 to 1, with 1 indicating highest wavelet coherence in both CWT results.

Additionally, the phase angle of the cross wavelet spectrum is introduced to represent the phase difference between two different time series (i.e., $a(t)$ and $b(t)$). It can be calculated from the real and imaginary parts of $W^{XY}$, whose graphical result is shown as an arrow. In the graph, the arrow pointing east (west) indicates the time series is positively (negatively) correlated in that point. Meanwhile, the arrow pointing south (north) indicates that the former time series (i.e., $a(t)$) precedes the latter one (i.e., $b(t)$) by a ratio of $\pi/2$, and vice versa. In general, the arrow is shown as a combination of horizontal and vertical directions. For example, an arrow pointing south-west direction indicates that the two different time series (i.e., $a(t)$ and $b(t)$) are negatively correlated with the former time series (i.e., $a(t)$) preceding the latter one (i.e., $b(t)$).

## 4. Results

### 4.1. Deformation Rate of Eastern Beijing Plain

Previous studies have shown that the deformation of our study area mainly occurred in the vertical direction (e.g., [14,15]). Therefore, in this work, it is reasonable to convert the line-of-sight (LOS) deformation into the vertical direction. The vertical deformation rate of the eastern Beijing plain was retrieved from C-band Sentinel-1 observations by using the abovementioned progressive small baseline subsets (SBAS) InSAR approach. It can be seen from Figure 4 that the land subsidence of the eastern Beijing plain was relatively severe and the annual average deformation velocity was −150 mm/yr~20 mm/yr from July 2015 to December 2021. Significant land subsidence regions were mainly concentrated in the eastern part of Chaoyang District (e.g., Jinzhan, Guanzhuang, Shuangqiao and Heizhuanghu) and the northwestern part of Tongzhou District (e.g., Songzhuang, Yongshun, Liyuan, Taihu and Lucheng). During the monitoring period, the maximum annual deformation velocity was about −150 mm/yr, which was located in the south of Jinzhan in Chaoyang District. Another noteworthy phenomenon was that the deformed regions of Chaoyang District and Tongzhou District developed to connect into a whole piece with wide spatial coverage.

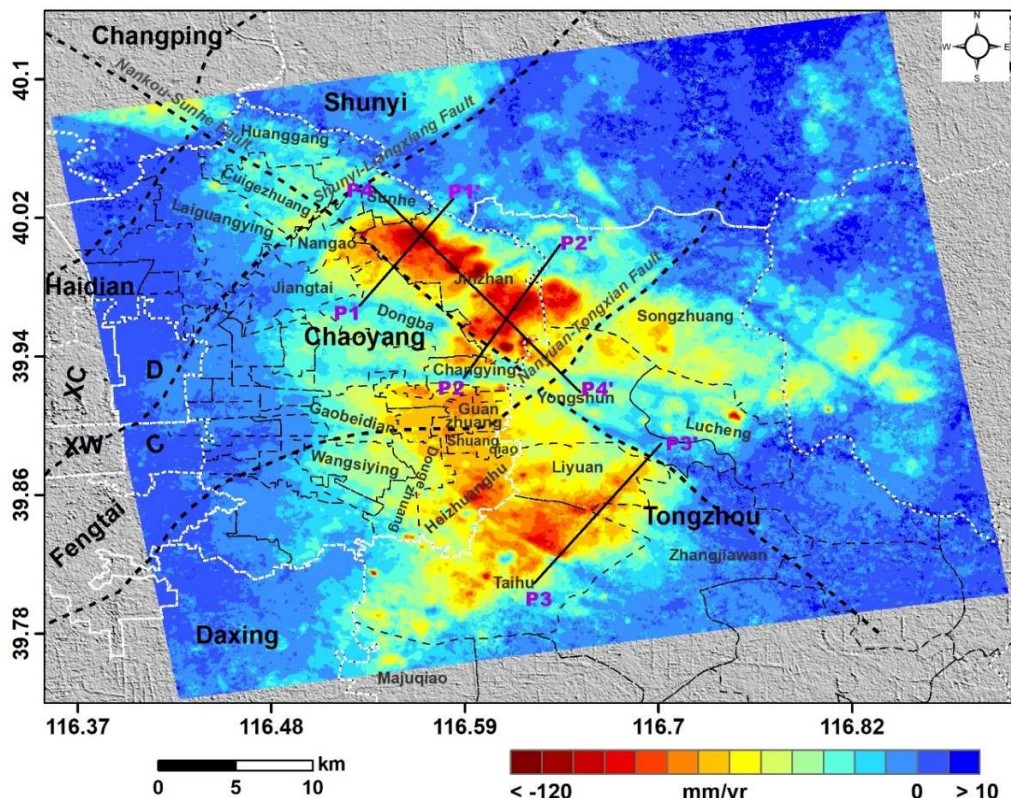

**Figure 4.** Annual average deformation rate map of eastern Beijing plain from July 2015 to December 2021 in the vertical direction. The positive values mean uplift, while the negative values mean subsidence. The thick black dashed line represents the main active faults, while the thick black solid line represents the location of the profiles across the faults.

*4.2. Time Series Deformation of Eastern Beijing Plain*

The cumulative time series deformation from 30 July 2015 to 13 December 2021 was also calculated in this study (Figure 5). It can be seen that obvious patterns were revealed in the spatial distribution, where the cumulative deformation magnitude and extent gradually increased with time in the eastern part of Chaoyang District and the northwestern part of Tongzhou District. From 30 July 2015 to 13 December 2021, the area with the cumulative deformation greater than −100 mm was about 651.1 km$^2$ and the maximum cumulative subsidence reached around −950 mm during the observation period, which was located in the south of Jinzhan in Chaoyang District.

In addition, the ground deformation at different stages from 2016 to 2021 were illustrated in Figure 6. From 2016 to 2018, the maximum deformation reduced from −192 mm to −173, while the areas with the deformation greater than −50 mm and −100 mm decreased from 332.52 km$^2$ to 249.66 km$^2$, and from 46.6 km$^2$ to 8.47 km$^2$, respectively. From 2019 to 2021, the maximum deformation reduced from −133 mm to −102 mm, while the areas with the deformation greater than −50 mm and −100 mm decreased from 63.06 km$^2$ to 14.64 km$^2$, and from 0.58 km$^2$ to 0.01 km$^2$, respectively. Then, the deformation evolution of the feature points (labeled as FP1~FP6 in Figure 6a) at the typical deformation centers were obtained and analyzed (Figure 7). From Figure 7, FP1 showed a maximum deformation of −148.6 mm in 2016 and then suddenly decreased to −79.1 mm in 2019. Subsequently, the deformation continued to slow down, with a minimum of −50.85 mm in 2021. FP2 also showed a maximum deformation of −170.94 mm in 2016, and then fell to −75.49 mm in 2020. Subsequently, the deformation increased slightly to −86.37 mm in 2021. FP3 reached the maximum deformation of −173.21 mm in 2017, and then gradually declined to −30.55 mm in 2021. For FP4 and FP5, the deformation trend was similar to that of FP3. For FP6, the deformation was relatively steady and maintained at −60~−80 mm,

except for a slight increase in 2019 and 2020. Generally speaking, the time series deformation of typical deformation centers showed that the land subsidence slowed down in 2017, which was about two years later than the start time of SNWD Project. Moreover, the corresponding complete time series of the feature points was shown in Figure 8. From Figures 7 and 8, it could be concluded that the area and magnitude of major deformation showed a significant decay trend from 2016 to 2021, even in the severe subsiding regions.

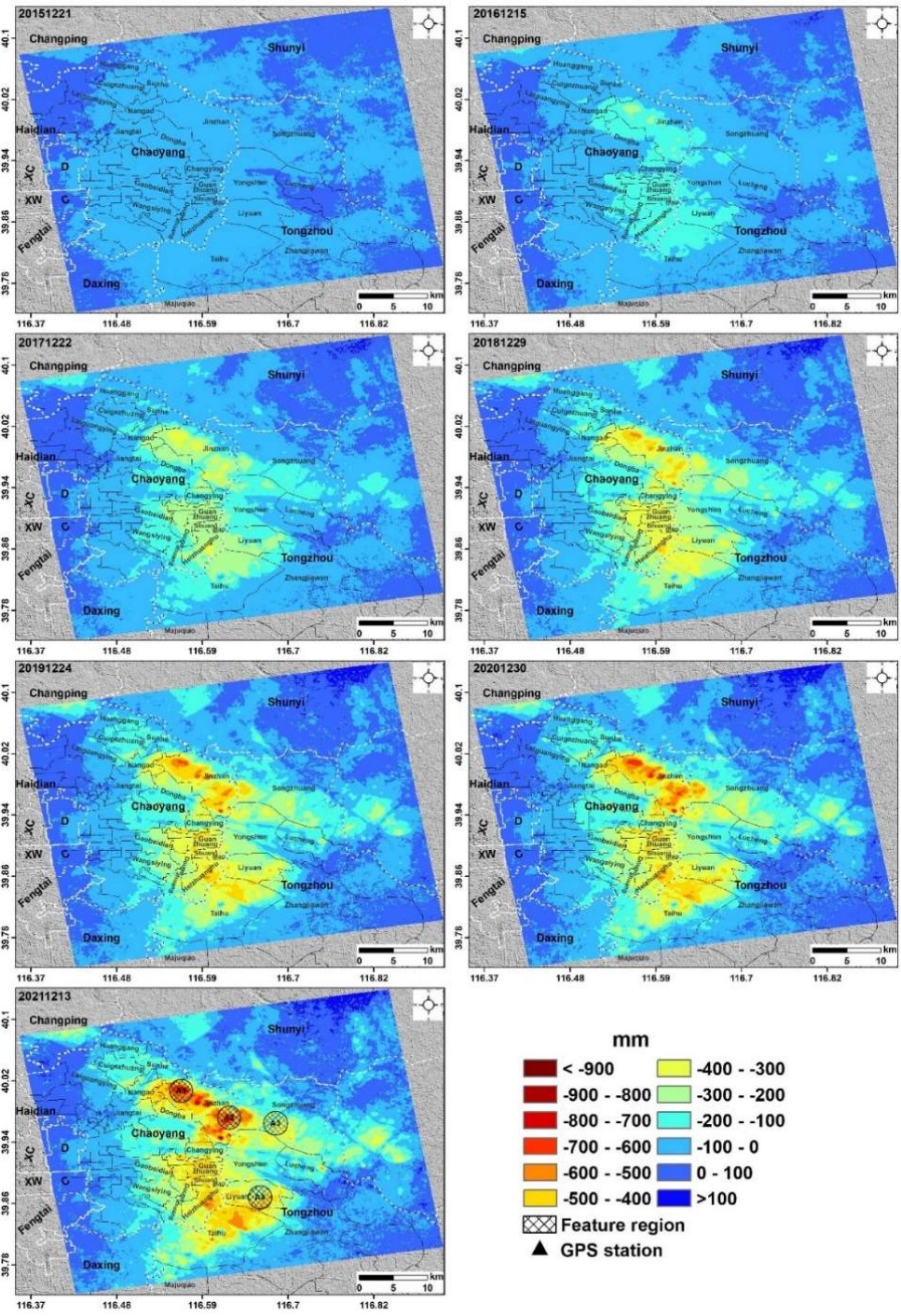

**Figure 5.** Cumulative vertical deformation from 30 July 2015 to 13 December 2021. The color bar indicates the vertical deformation in millimeters. The dashed white line is the county boundary of Beijing administrative district.

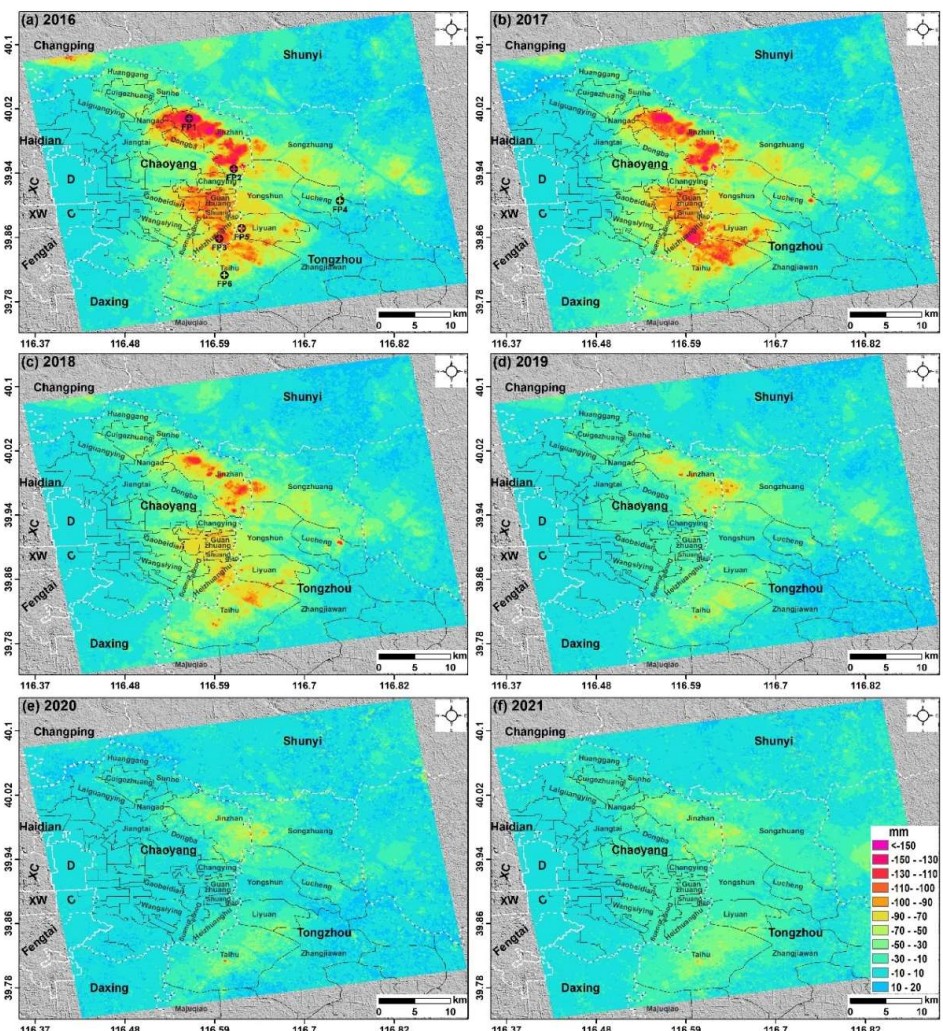

**Figure 6.** The distribution characteristics of land subsidence at different stages ((**a**) 2016, (**b**) 2017, (**c**) 2018, (**d**) 2019, (**e**) 2020, and (**f**) 2021) during the observation period in the eastern Beijing plain.

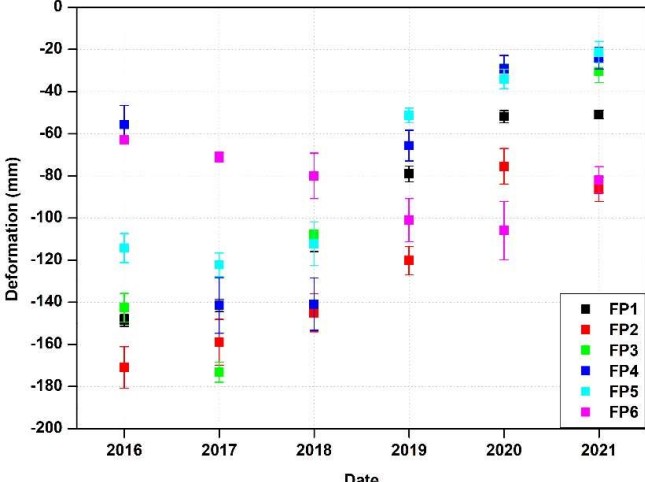

**Figure 7.** Annual Deformation of feature points located at typical deformed regions of Jinzhan (FP1~FP2), Heizhuanghu (FP3), Lucheng (FP4), Liyuan (FP5) and Taihu (FP6). Here the point targets within a 100-m radius buffer zone are selected to obtain the corresponding uncertainty of the InSAR-derived deformation. The error bar in each point represents the uncertainty of 1σ.

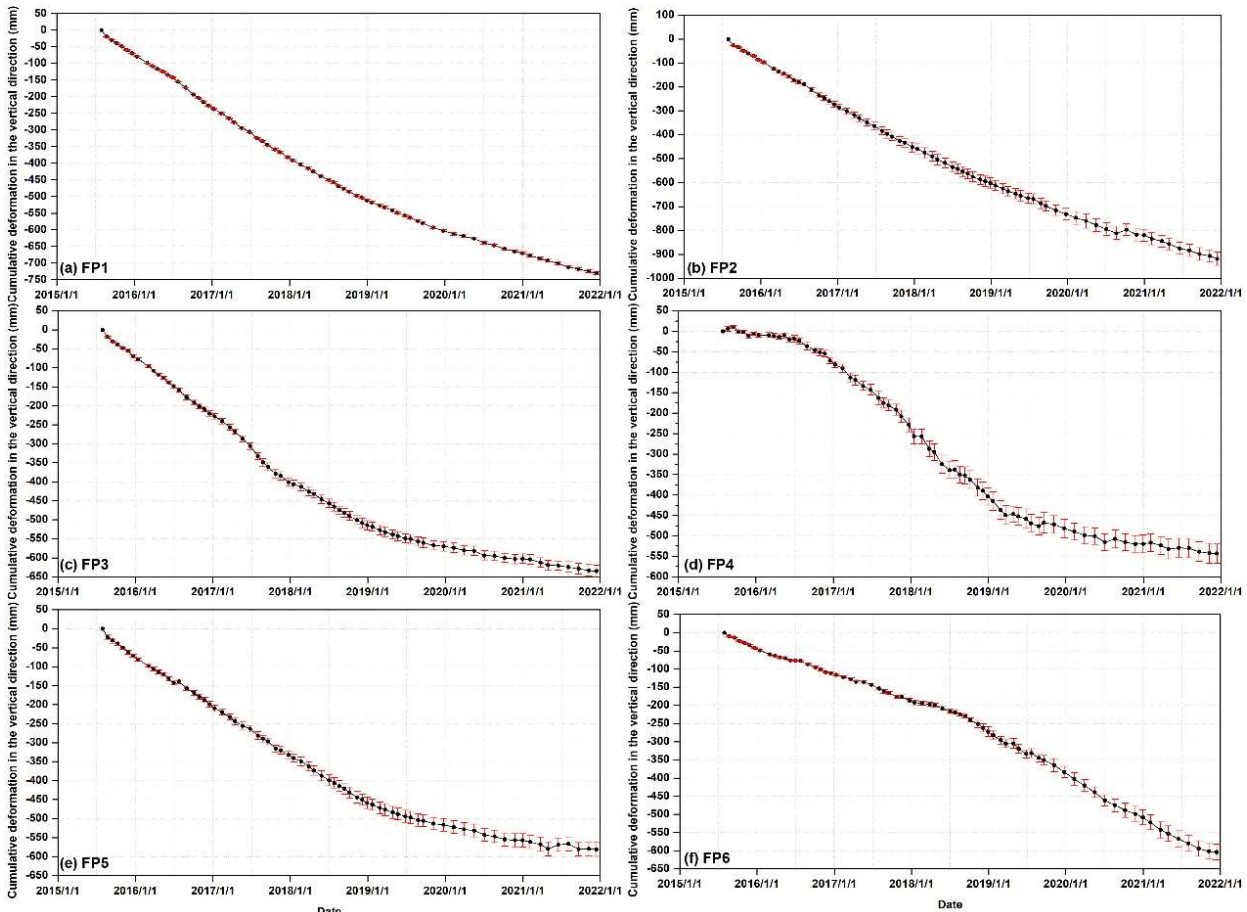

**Figure 8.** Time series deformation of feature points located at typical deformed regions of Jinzhan ((**a**) FP1~(**b**) FP2), Heizhuanghu ((**c**) FP3), Lucheng ((**d**) FP4), Liyuan ((**e**) FP5) and Taihu ((**f**) FP6).

### 4.3. Validation of InSAR Measurements

In order to reasonably validate the InSAR-derived deformation, we adopted two different validation strategies in this study. The first strategy referred to the full validation, which assessed the accuracy of InSAR-derived time series deformation with the continuous GPS observations. The second strategy referred to the trend validation, which conducted the comparison of deformation rates between InSAR and leveling measurements.

In this work, one GPS station and eighteen second-order leveling observations are collected, whose positions were shown in Figure 1. To reasonably compare the results from different techniques, we calibrated them into the common reference point. Then, we selected the coherent pixels within a 50-m radius buffer zone of each GPS station or leveling benchmark and averaged the corresponding InSAR-derived deformations around them. The cumulative time series deformation of the GPS station (shown as triangle in Figure 1) was presented in Figure 9, indicating that the InSAR-derived deformation agreed well with the GPS observations. The root mean square error (RMSE) between them was about 6.4 mm. A good agreement was also presented from the comparison result between InSAR observations and leveling measurements (Figure 10). The root mean square error (RMSE) and $R^2$ value (the confidence level at 95%) between them was about 4.3 mm and 0.99, respectively. The above comparison results suggested that the InSAR-derived deformation had a high accuracy and could provide reliable information for the following analysis on the potential influences of land subsidence in our study area.

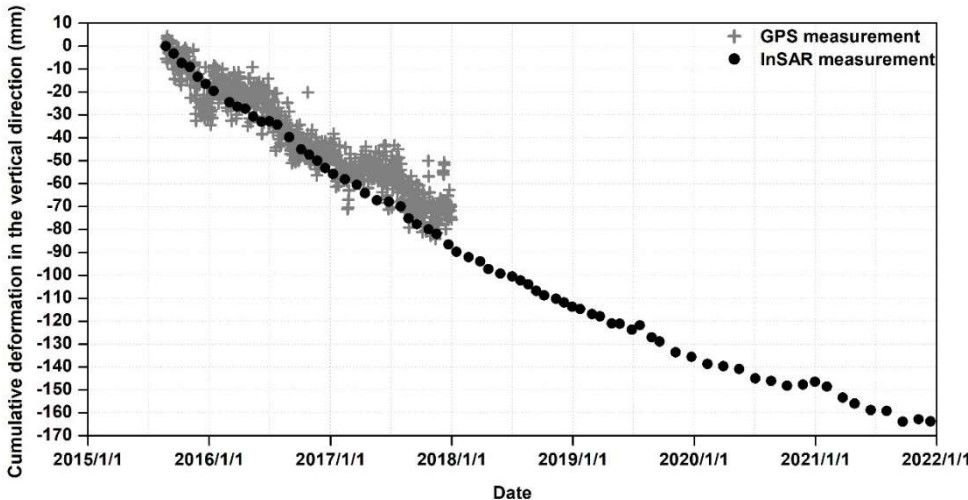

**Figure 9.** Cumulative time series deformation obtained from InSAR-derived and GPS observations.

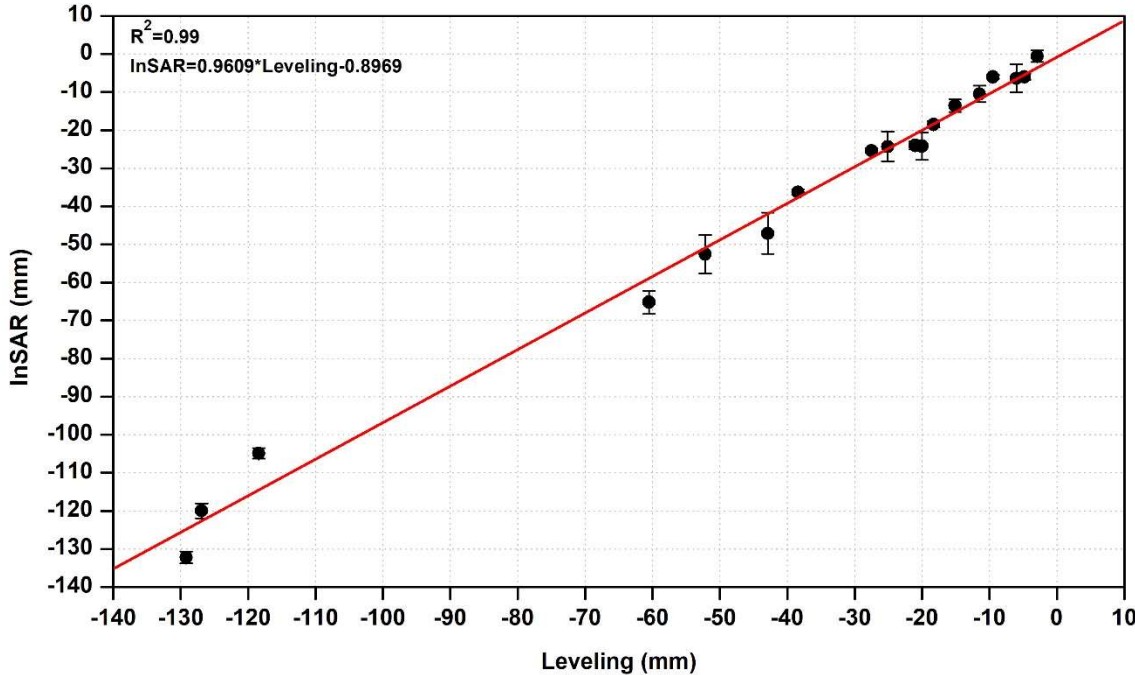

**Figure 10.** Comparison of InSAR-derived land subsidence and leveling measurements in the year of 2017. Here the point targets within a 50-m radius buffer zone are selected to obtain the corresponding uncertainty of the InSAR-derived deformation. The error bar in each point represents the uncertainty of 1σ.

## 5. Discussion

### 5.1. Cross Wavelet Transform (XWT) and Wavelet Transform Coherence (WTC) on Groundwater Level Change and InSAR-Derived Deformation

To reveal the correlations between the groundwater level change and ground deformation in the eastern Beijing plain, we conducted XWT and WTC on the groundwater level change and InSAR-derived deformation. Due to the lack of groundwater level information before June 2019, the correlation analysis was just conducted from June 2019 to December 2021 in this study. In addition, only the monthly groundwater level data could be collected, so XWT and WTC were applied based on the monthly groundwater level change and InSAR deformation change. Furthermore, considering the variation of groundwater exploitation in space, a buffer zone within the radius of 1.5 km was adopted to define the study area (A1~A4 shown in Figure 5). The mean value of each buffer zone

was adopted to represent the groundwater level change and InSAR deformation change of each defined study area. Figure 11 showed the time series of monthly groundwater level change and InSAR-derived deformation change of the selected regions, while Figure 12 represented the corresponding cross wavelet power spectrum and wavelet coherence spectrum. The XWT power spectrum showed the distribution of frequency component with time, while the WTC power described the coherence of the different time series in the time-frequency space. Arrows indicated the relative phase difference between groundwater level change and InSAR-derived deformation change. The thick contour designated the area that has passed the red noise test with confidence level at 95%, while the cone of influence (COI) was displayed as the thinner black line and the regions outside the COI were displayed as lighter shadows. From Figures 11a and 12a,e, at A1, the groundwater level change and InSAR-derived deformation change were basically in-phase correlation between June 2019 and December 2021 with the 10-month resonance period. If we converted the cross wavelet coherence into a lagging time, the InSAR-derived deformation change lagged behind the groundwater level change by 0~1 month. At A2, the groundwater level change and InSAR-derived deformation change were anti-phase correlation between May 2020 and September 2020, indicating that groundwater level change was not the main control factor of land subsidence. From July 2020 to October 2021, in-phase correlation was observed and the lagging time between InSAR-derived deformation change and groundwater level change was about 0~2 months (Figures 11b and 12b,f). The similar in-phase correlation and lagging time were observed at A3 (Figures 11c and 12c,g). However, the InSAR-derived deformation change preceded the groundwater level change from June 2019 to December 2019, indicating that the groundwater exploitation may not be the key influencing factor. At A4, in-phase correlation was clearly observed during the whole period with resonance period of 4~10 months and the lagging time of 0~3 months (Figures 11d and 12d,h)), revealing that the land subsidence was mainly controlled by the groundwater level change. Our results may provide guiding information on the lagging characteristic between potential influencing factors and the observed deformation [49], which is of great significance to reveal the deformation mechanism of regional subsidence and establish the hydrogeological models.

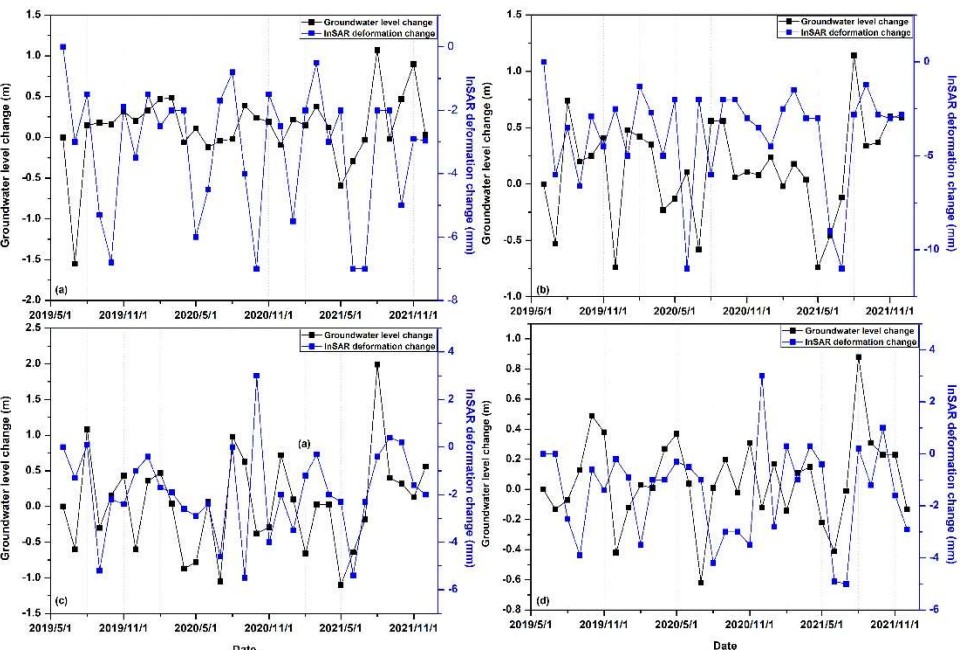

**Figure 11.** Time series of the monthly groundwater level change and InSAR-derived deformation change of the selected regions ((**a**) A1, (**b**) A2, (**c**) A3 and (**d**) A4).

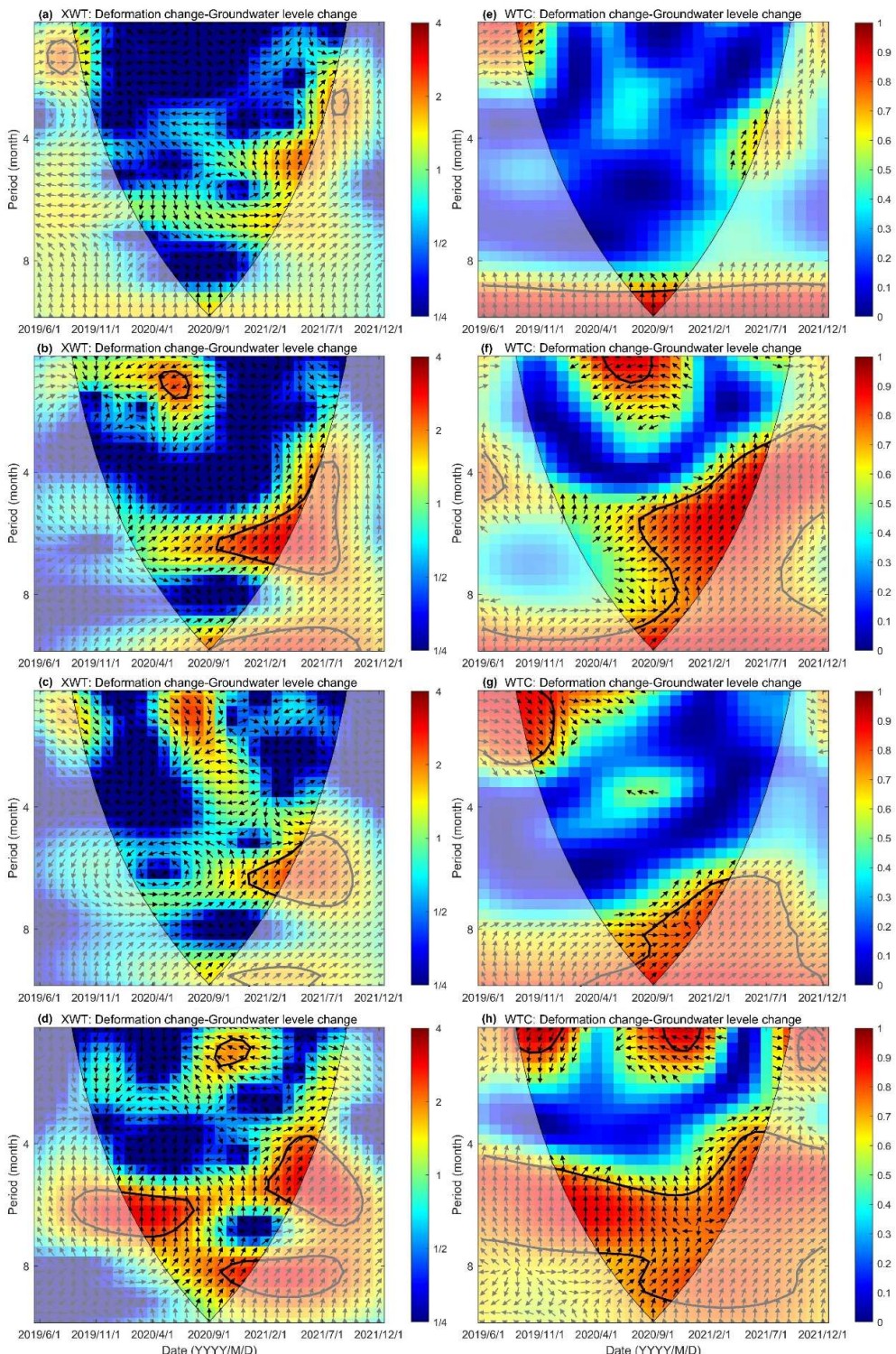

**Figure 12.** The cross wavelet power spectrum and the wavelet coherence between groundwater level change and InSAR-derived deformation change at four selected regions (A1: (**a**,**e**); A2: (**b**,**f**); A3: (**c**,**g**); A4: (**d**,**h**)). Pointing-right arrows: in-phase correlations, Pointing-left arrows: anti-phase correlations; Straight-down arrows: groundwater level change preceding InSAR-derived deformation change by 90°. The color bar of XWT and WTC shows the common cross-wavelet power of groundwater level change and InSAR-derived deformation change. The blue represents the low power while red represents the high power.

### 5.2. Correlation between Land Subsidence and Faults

Bai et al. [43] recorded that the compressible layer related to ground subsidence in the Beijing plain mainly occurred in the Quaternary sedimentary layer, and the sedimentary process was affected by the active geological faults. Therefore, the thickness of Quaternary sediments varied in spatial distribution, which provided the geological conditions to induce uneven ground subsidence. To reveal the correlation between ground subsidence and active geological faults, the annual average deformation rates along four profiles were analyzed (Figure 13), whose positions were marked in Figure 4. As shown in Figures 4 and 13, the boundaries of some subsiding regions corresponded to the traces of active geological faults, such as Nankou-Sunhe fault in the NW direction and Nanyuan-Tongxian fault in the NNE direction, which indicated that the spatial distribution of land subsidence was restricted by the active geological faults. Our results agreed well with previous studies [53]. In fact, many studies have reported similar phenomena in other regions, such as Mexico [4], Taiyuan [20], San Bernardino [54]. According to their opinions, the geological fault acts as a barrier to groundwater flow field, changing the horizontal hydraulic conductivity of aquifer systems and resulting in the differential groundwater level on both sides of the fault. This is the reason why the observed deformation rates are different on both sides of the Nankou-Sunhe fault and Nanyuan-Tongxian fault. The observed structural control of ground subsidence caused differential deformation rates on both sides of the fault, which may have an impact on urban infrastructures, for example, causing the damage to roads and railways. Furthermore, InSAR has proven to be an effective technique for mapping the active geological faults, which should be taken into consideration in future urban planning to avoid the damage of urban infrastructures owing to different deformations. At last, it is essential to conduct the continuous long-term deformation monitoring in these regions in order to update the kinematic data of geological faults and identify serious situations that could lead to the damage of urban infrastructures.

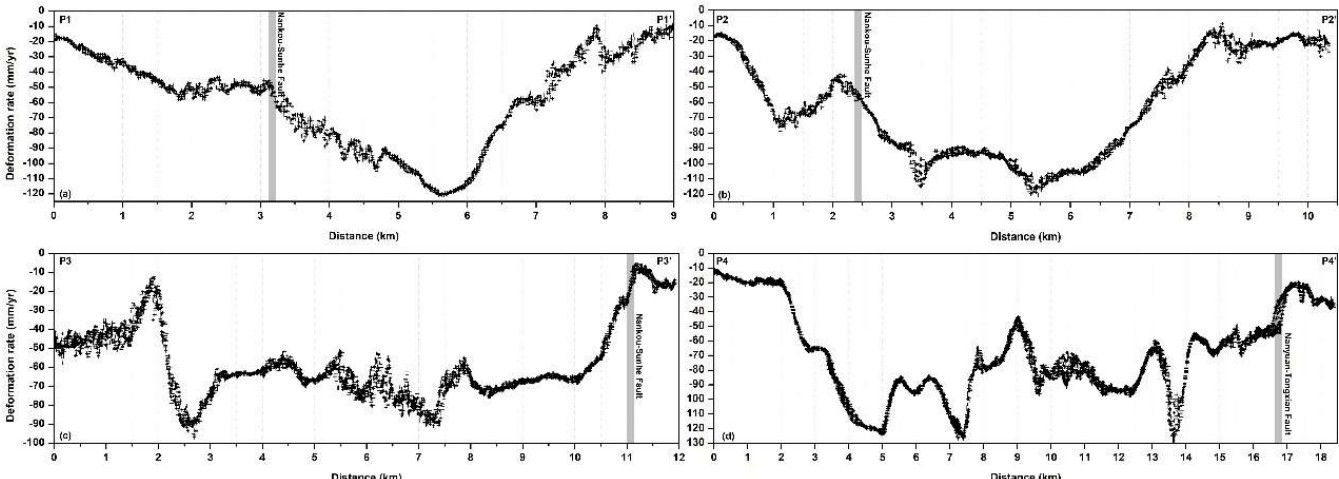

**Figure 13.** Vertical deformation rates along the profiles of P1P1′, P2P2′, P3P3′ and P4P4′. The locations of these profiles are shown in Figure 4. The gray bar shows the location of the active geological fault shown in Figure 4.

### 5.3. Correlation between Land Subsidence and Urban Expansion in Beijing's Sub-Administrative Center

Beijing's sub-administrative center (BSAC), located in the Tongzhou District, is of great significance for the construction and development of the Beijing-Tianjin-Hebei Metropolitan Area [55]. In general, the rapid urban expansion could aggravate the ground subsidence and threaten the safe operation of cities. Because the urban expansion was likely highly heterogeneous both in time and space, the potential relationship between ground subsidence and engineering construction was only revealed at the single- building scale in this study. Thus, a single building (labeled P) was selected for the deformation

time series analysis, which belonged to a newly built Central Business District in the BSAC (Figure 14). The location of building P was shown in Figure 1. The cumulative deformation from 2015 to 2021 was shown in Figure 14a, which revealed that the time-varying deformation rates were clearly observed at point P during the whole monitoring period. The deformation of the building P was mainly characterized by two different stages: rapid settlement stage (T1) with the average deformation rate of −66.9 mm/yr and relatively stable stage (T2) with the deformation rate of −13.38 mm/yr after the completion of engineering construction. In order to further explain the deformation characteristics of the building P during different stages, the geological drilling information was described in Figure 14b, which showed that there was an aquifer filled with gravel and silt-fine sand below the building P. The total thickness of the aquifer was 13.5 m, where the effective stress on soil could increase rapidly and then the gradual compression of sediments occurred accordingly. During the process of foundation pit dewatering of building P, the pore pressure disappeared quickly. After the completion of foundation pit drainage, the groundwater level gradually rose. With the increase of pore water pressure, the effective stress on soil slowed down, leading to the decrease of deformation rate and even rebound. Previous studies showed that the compressive modulus of silty clay was approximately 8.64~11.41 and that of silt was about 9.71~17.22 in this area [55], indicating that the compressibility of silty clay and silt was high. When the effective stress increased gradually, the deformation characteristic of building P was established during the stage of T2. Based on the effective stress principle proposed by Terzaghi [56], once the pore water pressure disappears completely, the effective stress is equal to the total stress. Subsequently, the main consolidation of the soil will be finished, but it may last for many years. According to the collected Google Earth images shown in Figure 14c, it could be concluded that urban construction may be one of the influencing factor affecting the land subsidence in this area to some extent.

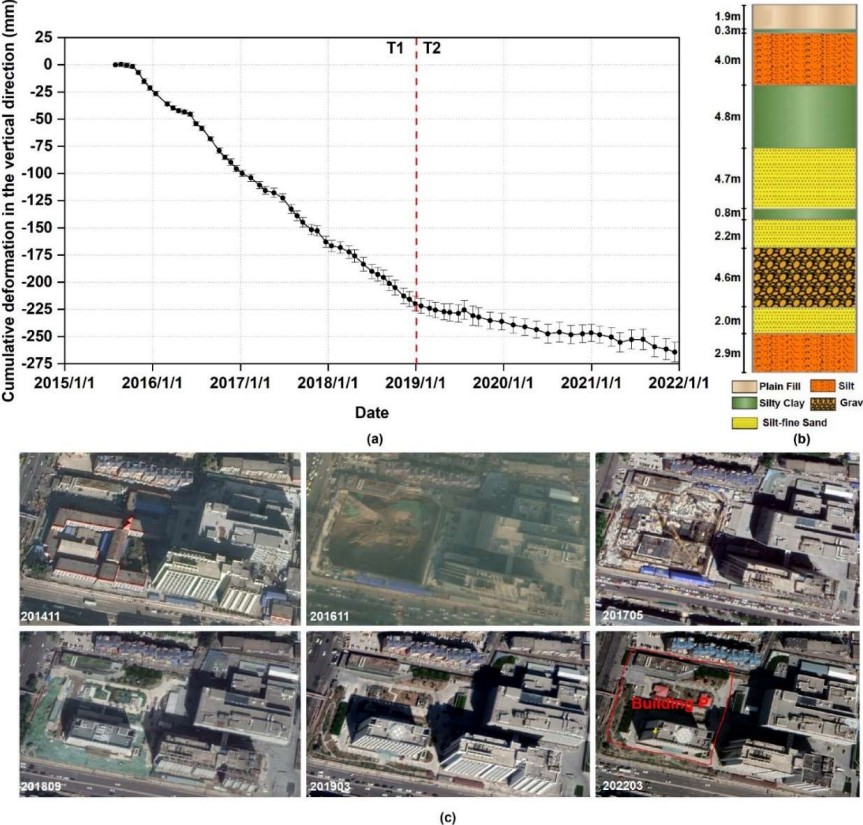

**Figure 14.** Cumulative deformation (**a**) and the geological drilling information of Building P (**b**), and (**c**) several typical enlarged optical remote sensing images obtained from Google earth of Building P.

Based on the above analysis, the InSAR-derived ground subsidence in the eastern Beijing plain is mainly induced by the excessive exploitation of groundwater, while its spatial distribution and extension is restricted by the local geological faults. In general, under the implementation of the SNWD Project, the excessive exploitation of groundwater in the eastern Beijing plain has been restricted and alleviated to a certain degree. However, it is a long-term and challenging task for the prevention and control of land subsidence caused by excessive exploitation of groundwater. It is necessary to wait for a long time to assess the role of the SNWD Project in replacing partial water demand and slowing down the excessive exploitation of groundwater in the capital. In addition, the urban construction (e.g., new buildings) may be one of the influencing factors affecting the ground subsidence to some extent. Therefore, in order to better study the spatiotemporal evolutions and influencing factors of land subsidence before and after the SNWD Project, we will adopt multi-source (e.g., Sentinel-1A/B, TerraSAR-X, ALOS PALSAR-2) or multi-track (e.g., ascending-track, descending-track) SAR data to retrieve the multi-dimensional deformations, combine optical remote sensing images to explore more influencing factors (e.g., floor areas of building, precipitation, land use type, urban expansion at the regional scale) on land subsidence and predict the future deformation (e.g., the deformation in 2050) with machine learning algorithms in the future research.

## 6. Conclusions

In this study, the deformation after the South-to-North Water Diversion (SNWD) Project in the eastern Beijing plain was retrieved by using the progressive small baseline subsets (SBAS) InSAR approach using Sentinel-1 SAR observations obtained from July 2015 to December 2021. The results showed that the eastern Beijing plain was still suffering severe ground subsidence and the maximum deformation rate was about $-150$ mm/yr in the vertical direction, which agreed well with the results published in the previous studies (e.g., $-146$ mm/yr [11]). Two main subsiding bowls with the deformation rate more than $-100$ mm/yr were identified in the eastern part of Chaoyang District (namely Jinzhan and Guanzhuan-Heizhuanghu) and the northwestern part of Tongzhou District (namely Songzhuang-Yongshun-Liyuan-Taihu). The InSAR-derived time series deformation agreed well with that from GPS observations and the RMSE was about 6.4 mm. Additionally, the validation between InSAR-derived deformation and leveling measurements indicated a high agreement, with the RMSE of 4.3 mm and the $R^2$ value up to 0.99 (the confidence level at 95%). Then, the deformation of six typical feature points at different stages revealed that the ground subsidence slowed down in 2017, which was about two years later than the start time of SNWD Project. Furthermore, the results obtained from cross wavelet transforms and wavelet transform coherence on the groundwater level change and InSAR-derived deformation change showed that the dynamic variation of groundwater fluctuation was the main influencing factor in the severe subsiding zones. The groundwater level change preceded land subsidence by 0~2 months in the eastern Beijing plain. In addition, the urban construction (e.g., new buildings) may be considered as one of the influencing factor on the ground subsidence to some extent.

This research provides important information about the latest land subsidence in the eastern Beijing plain, which may serve the decision-making in reducing the subsidence-related disasters and the management of water resources in Beijing.

**Author Contributions:** Conceptualization, Y.L. and Z.L.; methodology, Y.L.; software, X.Y.; validation, Y.L. and X.Y.; formal analysis, Y.L.; investigation, X.Y.; resources, Y.L.; writing—original draft preparation, Y.L. and X.Y.; writing—review and editing, M.Y. and B.L.; visualization, Y.X. and B.L.; funding acquisition, Y.L., Y.X. and B.L. All authors have read and agreed to the published version of the manuscript.

**Funding:** This work was supported by the National Natural Science Foundation of China (grant numbers 42104030, 42174055, 41962018), the Jiangxi Natural Science Foundation (grant numbers 20202BABL212016, 20212BAB204003), Key Laboratory for Digital Land and Resources of Jiangxi Province, East China University of Technology (grant number DLLJ202013) and the Doctoral Scientific Research Foundation of East China University of Technology (grant number DHBK2018004).

**Data Availability Statement:** Not applicable.

**Conflicts of Interest:** The authors declare no conflict of interest.

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
