# Peer review of "Characterizing Spatiotemporal Patterns of Land Subsidence after the South-to-North Water Diversion Project Based on Sentinel-1 InSAR Observations in the Eastern Beijing Plain"

_remotesensing, doi:10.3390/rs14225810_

Round 1

Reviewer 1 Report

General comments:

·      When the only type of surface deformation is subsidence, that term should be used, since we are only discussing subsidence in this paper, just stick to call all surface deformation as subsidence.

·      I appreciate the validation portion of the paper, good science.

Line 13 “The eastern Beijing plain has been suffering severe land subsidence for the last decades”

Line 14 “long-term excessive extraction of groundwater resource” The term of ground water over draft.

Line 23 Not necessarily a shortage, again an overdraft issue due to a net balance deficit

Line 63-64 “which has shown that land subsidence has been developinged in the eastern Beijing plain for the last decades”

Line 76-85: “The main objective of this work is to reveal the latest spatiotemporal changes of surface deformation over the eastern Beijing plain and assess the roles that groundwater exploitation, urban construction and geological faults played in the surface deformation. Firstly, a progressive SBAS-InSAR approach was adopted to obtain the current deformation rate and deformation time series in the eastern Beijing plain (2015-2021) by using 72 C-band Sentinel-1 SAR images. Then, we combined the InSAR-derived surface deformation and groundwater level measurements to reveal the potential relationship and lagging time between the InSAR-derived surface deformation and groundwater changes. Finally, the possible links between the InSAR-derived surface deformation, urban expansion and geological faults are discussed.”

Reviewer 2 Report

General comments: 

In this paper, the spatiotemporal patterns of surface deformation in the eastern Beijing plain are analyzed based on Sentinel-1A InSAR using progressive SBAS-InSAR technique. The results of the study are reasonable, worth publishing. 

Detailed comments are as follows. I hope my comments will help to improve the manuscript. 

Specific comments:

 1. Introduction

Line 61-67

The authors mentioned that there are several related research on land subsidence in Beijing plain, showing large deformation rate.

Compared with previous researches, are there any improvements in data or methods of your study? Is there any similarity between previous study and yours? Or you can give a brief summary of the methods used in previous studies to highlight the innovation of your method.

Line 71

References needs to be added for the statement of ‘…changing the status of land subsidence.’

Line 75

‘?’ changed to ‘.’

 2. Study area and dataset

Line 88

‘16412’ changed to ’16,412’ and similar issues need to be checked through the full text.

Line 125-126

What is the basis for setting the spatial and temporal baseline thresholds?

 4. Results

Line 230

Why is there a ground uplift? There is a large area of ground uplift in Figure 4, although the amount is small. Is it caused by a man-made object, or by passive earth pressure, or something else.

Figure 5

It is recommended to add the north pointer and scale for the figure.

Figure 5

The font of latitude, longitude and place names in the picture should be enlarged for easy reading.

Line 318-319

The sources of GNSS observation and leveling measurements need to be stated where appropriate.

5. Discussion

The discussion about the limitation of your study and its future prospects can be added at the end of this section as a separate paragraph.

 6. Conclusions

The conclusion should be a relatively refined statement. Some statements are future perspectives and can be placed in the discussion section.
